# Analysis of Melanoma Gene Expression Signatures at the Single-Cell Level Uncovers 45-Gene Signature Related to Prognosis

**DOI:** 10.3390/biomedicines10071478

**Published:** 2022-06-22

**Authors:** Mohamed Nabil Bakr, Haruko Takahashi, Yutaka Kikuchi

**Affiliations:** 1Department of Biological Science, Graduate School of Science, Hiroshima University, Kagamiyama 1-3-1, Higashi-Hiroshima, Hiroshima 739-8526, Japan; d180893@hiroshima-u.ac.jp; 2National Institute of Oceanography and Fisheries (NIOF), Cairo 11516, Egypt; 3Graduate School of Integrated Sciences for Life, Hiroshima University, Kagamiyama 1-3-1, Higashi-Hiroshima, Hiroshima 739-8526, Japan

**Keywords:** gene expression signatures, melanoma, single-cell RNA-sequencing, bulk RNA-sequencing, The Cancer Genome Atlas, prognostic signature

## Abstract

Since the current melanoma clinicopathological staging system remains restricted to predicting survival outcomes, establishing precise prognostic targets is needed. Here, we used gene expression signature (GES) classification and Cox regression analyses to biologically characterize melanoma cells at the single-cell level and construct a prognosis-related gene signature for melanoma. By analyzing publicly available scRNA-seq data, we identified six distinct GESs (named: “Anti-apoptosis”, “Immune cell interactions”, “Melanogenesis”, “Ribosomal biogenesis”, “Extracellular structure organization”, and “Epithelial-Mesenchymal Transition (EMT)”). We verified these GESs in the bulk RNA-seq data of patients with skin cutaneous melanoma (SKCM) from The Cancer Genome Atlas (TCGA). Four GESs (“Immune cell interactions”, “Melanogenesis”, “Ribosomal biogenesis”, and “Extracellular structure organization”) were significantly correlated with prognosis (*p* = 1.08 × 10^−5^, *p* = 0.042, *p* = 0.001, and *p* = 0.031, respectively). We identified a prognostic signature of melanoma composed of 45 genes (MPS_45). MPS_45 was validated in TCGA-SKCM (HR = 1.82, *p* = 9.08 × 10^−6^) and three other melanoma datasets (GSE65904: HR = 1.73, *p* = 0.006; GSE19234: HR = 3.83, *p* = 0.002; and GSE53118: HR = 1.85, *p* = 0.037). MPS_45 was independently associated with survival (*p* = 0.002) and was proved to have a high potential for predicting prognosis in melanoma patients.

## 1. Introduction

Melanoma is one of the deadliest and most aggressive types of skin cancer and is caused by the malignant transformation of melanocytes [1,2]. Melanogenesis, a melanin biosynthesis process, is primarily stimulated by ultraviolet radiation (UVR) and begins with the transformation of tyrosine to dopaquinone in a process catalyzed by tyrosinase [3,4]. Melanin pigment and melanogenesis have been proved to play a defensive role against UVR-induced skin cancer, including melanoma. On the other hand, based on the physicochemical properties of melanin and under uncontrolled melanogenesis conditions, melanin contributes to the malignant transformation of melanocytes [1,3,4,5]. This paradoxical action of melanin and melanogenesis in melanoma has been reviewed extensively [5]. Melanogenesis was shown to regulate melanoma metabolism [6] through stimulating HIF-1(-dependent and -independent) pathways and upregulating HIF-1α, GLUT-1, and stress-related genes [7]. Melanogenesis intermediates such as L-dihydroxyphenylalanine (L-DOPA) inhibit lymphocyte proliferation and generate an immunosuppressive environment resulting in drug resistance in melanoma cells [8,9]. Furthermore, a high level of melanin accumulation was associated with chemo-, immuno-, and radiotherapy resistance [9,10], and poor prognosis in patients with melanotic metastatic melanoma [10,11].

Although invasive melanoma accounts for less than 1% of skin cancers, it is responsible for 70% of skin cancer mortalities [12]. The introduction of a new immunotherapeutic strategy has improved the survival rate of patients with melanoma; however, tumor heterogeneity remains a major therapeutic problem [12,13,14,15]. Aggressive heterogeneous features are considered essential for invasion, metastasis, and drug resistance in melanoma [14,15]. Therefore, an understanding of the biological state of melanoma cells and the identification of new prognostic targets are needed. Melanoma staging was defined using three categories (T category = primary tumor, M category = distant metastases, and N category = regional lymph nodes), based on the American Joint Committee on Cancer (AJCC) criteria [16]. Recent advancements in transcriptome analysis have introduced the gene expression signature (GES) approach to better characterize tumors and define new prognostic targets [17,18,19].

GES is defined as the pattern of gene expressions that represent a biological state [20,21]. Hierarchical clustering for the bulk RNA-seq data of metastatic melanomas revealed four-class GESs (“high-immune”, “normal-like”, “pigmentation”, and “proliferative”) associated with patient survival regardless of the AJCC stage [22]. The same GESs were retained in primary melanomas [23] and assembled into high- and low-grade classes, indicating tumor aggressiveness and overall survival [23]. Analysis of the gene expressions of 329 melanoma patients from TCGA database, yielded three-class GESs (“immune”, “keratin”, and “MITF-low”) with the poorest survival of patients with the “keratin” signature [18]. Another study reported six-class GESs derived from the gene expression data of 687 patients with primary melanoma [24], which partially overlapped with the aforementioned four- and three-class GES studies [18,22]. Based on such investigations, various prognostic gene expression profiling tests have been introduced for clinical use [25]. Gerami et al. developed a prognostic gene signature composed of 31 genes to predict the metastatic risk in patients with cutaneous melanoma [26]. Thakur et al. reported a 150-gene prognostic signature to determine prognosis in patients with stage I primary melanoma and those receiving immunotherapy [24]. A recent study identified a 121-gene signature (Cam_121) associated with metastasis and overall survival in patients with melanoma, which was negatively correlated with the infiltration of the immune cells [27]. The gene sets included in the previously identified melanoma GESs showed poor overlapping, possibly because of the use of different sequencing platforms and the interference of the microenvironmental cells with the tumor gene expression in the bulk RNA-seq data [27]. Therefore, a higher-resolution analysis would overcome these issues and provide more accurate gene signatures for malignant cells.

Recent technological advances have enabled omics analysis at the single-cell level, such as single-cell RNA sequencing (scRNA-seq) [28,29]. The scRNA-seq is a useful and powerful tool for identifying complicated and diverse cell types and states of tumor and microenvironmental cells and for inferring the cell–cell communication between tumor cells and their microenvironment [28,30]. Numerous scRNA-seq analyses have been performed using a wide variety of melanoma cell lines and patient samples [31]. These studies have focused on characterizing the aggressive heterogeneity of melanoma and drug-resistant tumor cells, as well as the tumor microenvironment [31,32,33]. In this study, we aimed to characterize the GESs of patients with melanoma at the single-cell level using publicly available scRNA-seq data (GSE115978). Furthermore, the identified single-cell-based GESs were verified by the bulk RNA-seq of SKCM patients obtained from the TCGA database. A 45-gene signature significantly related to prognosis was identified and validated in TCGA-SKCM and three other bulk RNA-seq datasets (GSE65904, GSE19234, and GSE53118) of melanoma patients.

## 2. Materials and Methods

### 2.1. scRNA-seq Data

In this study, processed scRNA-seq data and annotation tables were downloaded from the GEO database under the accession number (GSE115978). The GSE115978 dataset consisted of 7186 cells corresponding to 31 melanoma patients. The untreated samples were specified and investigated to eliminate unnecessary factors. We performed an extensive analysis on the scRNA-seq data of four primary and eleven untreated metastatic melanoma samples.

### 2.2. scRNA Sequencing Analysis

Separately, count matrices of the malignant cells from the selected samples were introduced into R (4.1.0) and converted to Seurat object using the Seurat package (4.0.4) [34]. Cells with fewer than 2500 genes (nFeature_RNA > 2500) and mitochondrial counts >5% were excluded, resulting in a total of 23,686 genes across 981 cells. Expression data were normalized using the LogNormalize method with a scale factor of 10,000. The FindVariableFeatures function was used to define the highly variable genes. After scaling, the identified genes were subjected to principal component analysis (PCA). The top five principal components were introduced to the FindClusters function with a resolution of 0.4 for clustering the cells through the embedded Louvain algorithm [35]. The same components were used in the uniform manifold approximation and projection (UMAP) dimensionality reduction analysis [36] to colocalize and visualize the resulting seven Louvain clusters in a two-dimensional UMAP plot.

### 2.3. Cluster Signatures and Enrichment Characteristics

Differentially expressed genes (DEGs) between melanoma clusters were identified using the FindAllMarkers function in the Seurat package based on the Wilcoxon rank-sum test with min.pct = 0.25 and logfc.threshold = 1. The top 10 markers were hierarchically clustered and visualized using a heatmap using the DoHeatmap function. The DEGs were used for Gene Ontology (GO) enrichment analysis. GO enrichment was performed using the clusterProfiler package (4.1.4) [37]. Gene set enrichment analysis (GSEA) for the cancer Hallmark gene sets (Hallmark GSEA) was conducted using the GSVA package (1.41.3) [38]. Cancer Hallmark gene sets were retrieved from the molecular signature database (MSigDB). Hallmark GSEA results were visualized using the pheatmap package (1.0.12) (https://cran.r-project.org/web/packages/pheatmap/index.html). Interferon signature genes were retrieved from the HALLMARK_INTERFERON_ALPHA_RESPONSE and HALLMARK_INTERFERON_GAMMA_RESPONSE gene sets, and the expression levels of these genes were investigated in clusters 5, 1, and 3. Cell cycle heterogeneity along the melanoma clusters was assessed based on the cell cycle markers [32] embedded in the Seurat package. Cell cycle scores were assigned based on the expression of G2/M- and S-phase markers using the CellCycleScoring function.

### 2.4. Cell–Cell Interaction Analysis

Immune cell expression matrices were processed using the Seurat package, and the immune cell subtypes were automatically classified using SingleR (1.7.1) [39]. Immune cells were annotated using the Monaco Immune Database in the Celldex package (1.3.0) [39]. The normalized expressions of immune and melanoma cells were merged into one Seurat object. The CellChat package (1.1.3) [40] was used to infer the communication between melanoma cells and immune cells based on ligand–receptor interactions.

### 2.5. Bulk RNA-seq Data

The RNA expression data and clinical features of 470 (103 primary melanoma samples and 367 metastatic melanoma) SKCM patients were obtained from the TCGA database using the TCGABiolinks package (2.21.8) [41]. HTSeq-Counts files were downloaded, and genes were assembled based on the GRCh38.p12 (hg38) genome reference. Other bulk RNA-seq datasets (GSE65904, GSE19234, and GSE53118) of patients with melanoma were downloaded using the GEOquery package (2.62.2) [42] for prognosis validation.

### 2.6. TCGA-SKCM Data Analysis

To explore and represent the identified scRNA melanoma clusters (GESs) in the TCGA-SKCM samples, we used the CIBERSORTx algorithm [43]. The bulk RNA-seq expression matrix was normalized to counts per million (CPM) and scaled by calculating the log_2_ values. Additionally, a signature matrix of the scRNA clusters was constructed based on DEGs with |Log2FC| > 1 and *p* < 0.05. Both matrices were introduced into the CIBERSORTx online tool, and the default parameters of the cell fractions analysis module were used to deconvolute the bulk RNA-seq data. We used the xCell package (1.1.0) [44] to determine the enriched cell types within the identified GESs in TCGA-SKCM samples. We compared our identified GESs to previously reported transcriptomic signatures [18,22,23] based on the expressions of the representative genes of each signature. Additionally, we used the ComplexUpset package (1.3.3) [45] to visualize the intersections of the samples.

### 2.7. Survival Analyses

We classified TCGA-SKCM patients into six groups based on the dominant GES. The normalized gene expression matrix was used for overall survival analysis. The Surv function from the Survival package (3.2-13) (https://cran.r-project.org/web/packages/survival/index.html) was used to create a survival object based on TCGA-SKCM clinical data. We then fitted the survival curves for the different GES groups or for each GES group individually using the survfit function. The log-rank test was used, and we considered *p* < 0.05 as the threshold for significance. Kaplan–Meier survival curves were constructed using the ggsurvplot function from the Survminer package (0.4.9) (https://cran.r-project.org/web/packages/survminer/index.html). To compare the vital status within 4 years for patients of each GES group, we categorized TCGA-SKCM patients into dead (patients who died within the first 4 years) and alive (patients who were alive for more than 4 years) based on “days to death” and “days to last follow-up”, respectively.

### 2.8. Cox Regression Analyses

DEGs of cluster 5 (“Immune cell interactions” GES; |Log2FC| > 1.5 and *p* < 0.05) and cluster 4 (“Ribosomal biogenesis” GES; |Log2FC| > 1 and *p* < 0.05) obtained from the scRNA-seq analysis were screened to identify new therapeutic targets for melanoma. We used the clinicopathologic and transcriptomic data of TCGA-SKCM patients to perform univariate Cox regression analyses to detect the correlation between each DEG and the overall survival using coxph function from the Survival package (3.2-13). DEGs derived from the (“Ribosomal biogenesis” GES and correlated to prognosis (hazard ratio (HR) > 1 and *p* < 0.05) were selected and fitted by the ridge regression Cox model (alpha = 0) using the glmnet package (4.1-3) [46]. A 45-gene prognostic signature (MPS_45) was developed, and to validate the correlation between MPS_45 and prognosis, Cox regression analysis based on the gene expression scores was implemented. We used the singscore package (1.14.0) [47] to score the normalized gene expression matrices of TCGA-SKCM, GSE65904, GSE19234, and GSE53118 datasets against the 45 DEGs of MPS_45. After scoring, samples were grouped into “high” and “low” groups based on the median gene expression scores, log-rank tests were performed, and Kaplan–Meier survival curves were created. Furthermore, we constructed a time-dependent receiver operating characteristic (ROC) curve to evaluate the prognostic value of MPS_45 within 1, 3, and 5 years in the TCGA-SKCM patients using the pROC package (1.18.0) [48]. Finally, we performed multivariate Cox regression analyses to inspect the correlation between MPS_45 and clinicopathological features in melanoma survival. The results were visualized as forest plots using the forestmodel package (0.6.2) (https://cran.r-project.org/web/packages/forestmodel/index.html).

## 3. Results

### 3.1. Cell Clustering and GES Identification of Malignant Melanoma Cells

To investigate GESs in melanoma patients at the single-cell level, we used a public scRNA-seq dataset (GSE115978), in which cells from primary and metastatic sites of human melanoma patients were annotated [49]. Of the 31 patients with melanoma included in the GSE115978 dataset, 15 samples obtained from untreated patients were selected for further investigation (Appendix A). Based on rigorous quality control measures, 981 malignant melanoma cells with 23,686 features were obtained from the selected samples. We performed dimensionality reduction and unsupervised clustering for the 981 malignant melanoma cells using the UMAP algorithm without batch effect correction to clarify the information pertaining to each patient. The UMAP analysis revealed six distinct cellular clusters with a resolution of 0.4 (Figure 1A, clusters 0–5). To determine the appropriate resolution parameter for clustering the cells, we constructed a clustering tree [50] using different resolution values (0, 0.2, 0.4, 0.8, and 1.2). The clustering tree showed a region of stability at a resolution of 0.2–0.4 before clusters 0, 1, and 4 started to split (Figure 1B). Higher resolution values (0.8 and 1.2) caused over-clustering, with edges having a low proportion ratio and unstable clusters (Figure 1B). Furthermore, UMAP visualization based on patient origin showed that each cluster was derived from a single patient (Figure 1C). As each cluster consisted of the genetic profile of a single patient, individual patients were expected to have a specific gene profile.

To determine the gene signatures of the six scRNA melanoma clusters, we identified the DEGs in each cluster (Figure 1D and Appendix A) and performed GO enrichment (Biological Process) (Figure 2A and Appendix A) and the Hallmark GSEA (Figure 2B and Appendix A). The expression levels of the top 10 marker genes showed that the expression of these genes in each cluster was distinctive (Figure 1D), suggesting that each melanoma cluster has distinct gene signatures. GO enrichment and Hallmark GSEA revealed the representative enriched gene signatures in melanoma clusters as follows:

Cluster 0: GO enrichment analysis revealed that cluster 0 was mainly enriched with apoptosis- and proteolysis-related biological processes (Figure 2A). Several members of the MTRNR2L family genes (*MTRNR2L1*/*2*/*3*/*6*/*8*/*10*), which have anti-apoptotic activity [33,51], were highly expressed in cluster 0 (Appendix A). In addition, pro-apoptotic genes (*BAX*, *BAK1*, Procaspase-3 (*CASP3*), and Procaspase-9 (*CASP9*)) had the lowest expression levels in cluster 0 (Appendix A). Simultaneously, Hallmark GSEA showed that cluster 0 was exclusively enriched with the Hedgehog (Hh) signaling pathway (Figure 2B), possibly through the upregulation of *MYH9*, *NF1*, *CELSR1*, and *PTCH1* genes and the downregulation of *ETS2*, *NRCAM*, and *HEY1* genes (Appendix A).

Cluster 5: GO enrichment analysis showed that T cell activation and immune cell interactions were enriched as biological processes (Figure 2A), with high enrichment of Major Histocompatibility Complex (MHC) genes (Appendix A). In addition, Hallmark GSEA showed that allograft rejection, the inflammatory response, and interferon alpha and gamma responses were highly enriched in this cluster (Figure 2B). Based on these results, we employed cell–cell interaction analysis between melanoma and immune cells (Figure 3). After clustering immune cells into clusters of monocytes, dendritic cells, B cells, NK cells, T cells, CD8^+^ T cells, and CD4^+^ T cells (Figure 3A), we checked their interactions with the identified melanoma cell clusters (Figure 3B), especially with cluster 5 (Figure 3C–G). We found that cluster 5 mainly interacts with CD8^+^ and CD4^+^ T cells through MHC classes I and II pathways (Figure 3D–F). Notably, cluster 5 had the strongest cell–cell interactions with T cell subtypes compared with the other melanoma cell clusters (Figure 3D,E). Cluster 5 was the main source of ligands and the dominant mediator in the MHC-II signaling pathway (Figure 3E). Furthermore, we checked the type II interferon (IFN-II) signaling pathway and found strong interaction between cluster 5 and CD8^+^ cells (Figure 3G).

Cluster 1: GO enrichment analysis showed that in addition to T cell activation and immune cell interaction processes, pigmentation-related processes were enriched in this cluster (Figure 2A). Two melanogenesis-related genes, *TRPM1* and *TYRP1*, and two other pigmentation-related genes, *TPCN2* and *GSTP1*, were identified in the top 10 marker genes of cluster 2 (Appendix A). In addition, extra pigmentation-related genes (*DCT*, *EDNRB*, *MITF*, *GPR143*, *CDH3*, and *PMEL*) were highly expressed in this cluster (Appendix A). Hallmark GSEA showed that although the interferon alpha and gamma responses were highly enriched in this cluster, the expression profiles of interferon alpha and gamma response-related genes in cluster 1 were different from those in clusters 5 and 3 (Figure 2C). Consistent with this result, the interaction of CD8^+^/CD4^+^ T cells with cluster 1 was markedly reduced compared with that with cluster 5 (Figure 3). Furthermore, the enrichment of allograft rejection and inflammation-related genes was downregulated in this cluster compared with that in cluster 5 (Figure 2B).

Cluster 4: In this cluster, the expression of ribosomal proteins was very high (Appendix A), which is consistent with the enriched biological processes (cytoplasmic translation and ribosome biogenesis) found using GO enrichment analysis (Figure 2A). In addition, we found that the E2F targets, related to cell-cycle-related genes, were highly enriched in this cluster (Figure 2B), and that the ratio of S and G2/M phases in the cell cycle was higher in this cluster than in the other clusters (Appendix A).

Clusters 3 and 2: Based on the results of GO enrichment and Hallmark GSEA, both clusters 3 and 2 showed similar gene signatures (Figure 2A,B). Extracellular matrix organization (Figure 2A) and EMT (Figure 2B) were highly enriched in these clusters. Accordingly, we explored the expression of the EMT transcription factors (EMT-TFs) (*ZEB1*/*2*, *TWIST1*/*2*, *SNAI1*/*2*, *PRRX1* [52], and *FOXC2* [53]) in each cluster. The expressions of *ZEB2* and *SNAI2* fluctuated across the melanoma clusters (Appendix A). The expression levels of *ZEB1*, *SNAI1*, and *PRRX1* were higher in cluster 2 than in the other clusters (Appendix A). In contrast, the expression of *TWIST1* remained unchanged, whereas *TWIST2* and *FOXC2* were nearly undetectable (Appendix A).

Collectively, these results suggest six GESs for melanoma patients at the single-cell level (cluster 0: “Anti-apoptosis”, cluster 5: “Immune cell interactions”, cluster 1: “Melanogenesis”, cluster 4: “Ribosomal biogenesis”, cluster 3: “Extracellular structure organization”, and cluster 2: “EMT”) with distinctive genetic characteristics.

### 3.2. Identification of TCGA-SKCM Patients Expressing the Same GESs Determined Using scRNA-seq Analysis

Since each melanoma cell cluster (0, 5, 1, 4, 3, and 2) represented a definite GES in scRNA-seq, we next employed the CIBERSORTx algorithm to determine the fraction of melanoma cells with the same genetic characteristics in bulk RNA-seq data of the SKCM patients from TCGA database. We used the gene signatures of the identified melanoma scRNA-seq GESs to deconvolute 470 SKCM bulk RNA-seq samples from TCGA (Appendix A). TCGA-SKCM samples showed varied fractional representations of the scRNA melanoma GESs (Figure 4A). Based on the dominant representation of the scRNA melanoma GESs, we classified TCGA-SKCM samples into six groups (GESs) (Figure 4A). More than 87% of the bulk RNA-seq samples were classified into three dominant groups: “Immune cell interactions” (32.55%), “Ribosomal biogenesis” (27.45%), and “Extracellular structure organization” (27.45%) (Figure 4B). Furthermore, to confirm the classification of the TCGA-SKCM samples, Hallmark GSEA showed that each TCGA GES had an enrichment profile similar to that observed for each of the scRNA GESs (Figure 4C). Using cell-type enrichment analysis with xCell, we found that the “Immune cell interactions” GES was enriched with several types of immune cells (Figure 4D). On the contrary, immune cell enrichment was attenuated in the other GESs (Figure 4D).

A previous study used a hierarchal clustering analysis of 1,500 genes to classify 329 TCGA-SKCM samples into three subclasses (“immune”, “keratin”, and “MITF-low”) based on the gene functions [18]. In addition, another study reported four-class (“high-immune”, “normal-like”, “pigmentation”, and “proliferative”) GESs in 57 patients with stage IV melanoma [22]. These four-class GESs were further replicated and converged into two (“high-grade” and “low-grade”) GESs [23]. We compared the classification of the TCGA-SKCM samples between our identified GESs and previously reported signatures (Figure 5 and Appendix A). Our identified “Immune cell interactions” and “EMT” GESs showed high similarities with “immune” and “high-immune” GESs (Figure 5A) and “MITF-low” and “proliferative” GESs (Figure 5B), respectively. In line with this, the expression of the *MITF* gene showed the lowest level in our “EMT” GES (Figure 5C). Additionally, genes of the previously reported “high-grade” GES were strongly expressed in our “Melanogenesis” and “Ribosomal biogenesis” GESs (Appendix A). In contrast, the “low-grade” GES was highly represented in our “Immune cell interactions” GES (Appendix A). By comparing the overlap between TCGA-SKCM patients, we observed the highest overlap (54%) between the “immune” subclass and our “Immune cell interactions” GES (Figure 5D). However, the “keratin” subclass mainly overlapped with our “Ribosomal biogenesis” (28%) and “Extracellular structure organization” (25%) GESs (Figure 5D). These results suggest a partial overlap between our identified GESs and previously reported GESs.

### 3.3. GESs Were Correlated to Melanoma Prognosis

We explored the association between our identified GESs and survival outcome using log-rank tests and Kaplan–Meier survival analysis and found that TCGA-SKCM patients with “Melanogenesis”, “Ribosomal biogenesis”, and “Extracellular structure organization” GESs showed the poorest prognosis (*p* = 1.62 × 10^−5^) (Figure 6A). Consistent with this result, the survival rate within 4 years for patients having “Immune cell interactions”, “Anti-apoptosis”, and “EMT” GESs was more than 60% (Figure 6B). Furthermore, to investigate the relationship between each GES and the overall survival outcome independently, we compared the patients within each GES group with the remaining TCGA-SKCM patients (Figure 6C–H). Patients having “Anti-apoptosis” and “EMT” GESs were not correlated with prognosis (Figure 6C,H), which may be due to the few patients included in these GESs groups (5.74% and 2.98%, respectively) (Figure 4B). However, patients having “Immune cell interactions” GES showed significantly improved survival (*p* = 1.08 × 10^−5^) (Figure 6D), whereas patients having “Melanogenesis”, “Ribosomal biogenesis”, and “Extracellular structure organization” GESs showed significantly poorer prognosis (*p* = 0.042, *p* = 0.001, and *p* = 0.031, respectively) (Figure 6E–G).

### 3.4. Univariate Cox Regression Analyses for DEGs of “Immune Cell Interactions” and “Ribosomal Biogenesis” GESs

Given that “Immune cell interactions” and “Ribosomal biogenesis” GESs showed the highest significant positive (*p* = 1.08 × 10^−5^) and negative (*p* = 0.001) correlations, respectively, with survival rates in melanoma patients, we used the DEGs obtained from the scRNA-seq analysis and the survival data from TCGA-SKCM to explore the prognosis-related DEGs from each GES. Using univariate Cox regression analyses for the 56 DEGs of the “Immune cell interactions” GES (|Log2FC| > 1.5 and *p* < 0.05), we identified 49 DEGs (*ARHGDIB*, *C1QB*, *CCL5*, *CD14*/*4*/*53*/*69*/*74*/*8A*, *CORO1A*, *CTSS*, *CXCL10*, *CXCR4*, *GZMK*, *HCLS1*, *HLA*-*DPA1*/*DPB1*/*DQA1*/*DQB1*/*DRA*/*DRB1*/*DRB5*, *IFI30*, *IGLL1*/*5*, *IL32*, *IRF8*, *ITGB2*, *LAPTM5*, *LCP1*, *LSP1*, *LYZ*, *MS4A1*, *NKG7*, *PIK3IP1*, *PRF1*, *PTPN6*/*RC*, *RAC2*, *SASH3*, *SELL*, *SLA*, *SRGN*, *STK17B*, *TAGAP*, *TCIRG1*, *TMSB4X*, *TNFRSF1B*, and *UCP2*) related to improved survival (*p* < 0.05 and HR < 1) (Appendix A). For the “Ribosomal biogenesis” GES, out of 161 DEGs (|Log2FC| > 1 and *p* < 0.05), 49 DEGs were significantly correlated with prognosis (*p* < 0.05) (Figure 7). These included 45 DEGs (*ACTG1*, *AURKA*, *CCNB2*, *CDC20*/*6*/*A4*, *CENPF*, *CKS1B*, *COX6C*, *EIF3K*, *ESRP1*, *IMPDH2*, *KIF2C*, *KPNA2*, *LDHB*, *LSM4*, *MAD2L1*, *MCM4*/*7*, *MLANA*, *MRPL37*, *NCL*, *NME2*, *PRC1*, *PSMD8*, *RAN*, *RPL10*/*18*, *RPS19*/*3*, *RUVBL2*, *SAE1*, *SLC25A5*, *TIMM13*/*50*, *TK1*, *TOMM22*, *TPX2*, *TUBA1B*/*A1C*/*B4B*/*B6*, *TYRP1*, *UBE2C*, and *ZWINT*) with HR > 1 and 4 DEGs (*HMGB2*, *EMC2*, *COPS5*, and *ARHGDIB*) with HR < 1 (Figure 7).

### 3.5. Exploration and Validation of Melanoma Prognostic Signature

To develop a melanoma prognostic signature, DEGs derived from the “Ribosomal biogenesis” GES and potentially related to poor prognosis (HR > 1 and *p* < 0.05) (Figure 7) were further fitted by the ridge regression Cox model (Figure 8A). Univariate and ridge regression Cox analyses identified a possible melanoma prognostic signature composed of 45 DEGs (Appendix A), the elevated expression of which was linked to a worse prognosis (Figure 7) (referred to as MPS_45). TCGA-SKCM samples were divided into high and low MPS_45 groups based on the median gene expression scores of MPS_45. Cox regression analysis using the log-rank test showed a significant correlation (HR = 1.82, *p* = 9.08 × 10^−6^) between the MPS_45 and overall survival (Figure 8B). The same results were obtained after validating the prognostic characteristics of MPS_45 in three additional bulk RNA-seq melanoma datasets (GSE65904: HR = 1.73, *p* = 0.006; GSE19234: HR = 3.83, *p* = 0.002; and GSE53118: HR = 1.85, *p* = 0.037) (Figure 8C–E). To test the prognostic performance of MPS_45, we constructed time-dependent ROC curves using TCGA-SKCM samples (Figure 8F). The area under the curve (AUC) values for the 1-, 3-, and 5-year cohorts were 0.896, 0.779, and 0.798, respectively (Figure 8F). Multivariate Cox regression analyses showed that MPS_45 remained associated with poor prognosis when combined with the clinicopathological features of patients with melanoma (*p* = 0.002) (Figure 9A). Exploration of the MPS_45 genes in the TCGA-SKCM patients showed high expression levels in patients with “Extracellular structure organization”, “Melanogenesis”, and “Ribosomal biogenesis” GESs (Figure 9B). Furthermore, multivariate Cox regression analyses revealed that high MPS_45 scores (*p* < 0.001) and having these three GESs (*p* < 0.001, *p* = 0.002, and *p* < 0.001, respectively) jointly affected the survival of TCGA-SKCM patients (Figure 9C).

## 4. Discussion

Despite recent improvements in melanoma patient survival rates, aggressive heterogeneous molecular signatures [14,15,54] remain a serious issue. Therefore, better knowledge of the biology of melanoma cells and the identification of new prognostic targets are required. GESs have been proven to provide biological and prognostic information for patients with melanoma [18,22,23,24]. In this study, we clustered single melanoma cells using PCA and UMAP visualization, and each cluster was characterized based on enriched biological processes and pathways to identify six distinctive GESs of melanoma at the single-cell level.

Two (“Immune cell interactions” and “EMT”) out of the six identified GESs showed high similarities (Figure 5 and Appendix A) with previously identified melanoma signatures derived from bulk RNA-seq analysis [18,22]. The “Immune cell interactions” GES was characterized by a similarly high expression of MHC (Ⅰ and Ⅱ) genes as the previously reported “high-immune” GES [22]. Normally, cancer cells suppress the expression of MHC class Ⅰ and show reduced interaction with CD8^+^ cytotoxic T cells (CTL) to evade immune cells [55]. In contrast, in cells with “Immune cell interactions” GES, the expression of MHC class Ⅰ is increased, which predicts a high interaction with CTL. Taking advantage of our scRNA-seq analysis, we illustrated strong interactions of melanoma cells having “Immune cell interactions” GES with CD4^+^ and CD8^+^ T cells (Figure 3). This interaction may explain the improved survival of patients with immune signatures in our study and previous studies [18,22,24,56]. We identified an “EMT” GES in which the representative genes of previously identified signatures (“MITF-low” [18] and “proliferative” [22]) were highly expressed. In line with “MITF-low” [18] and “proliferative” [22] GESs, melanoma cells with “EMT” GES had the lowest level of *MITF* expression (Figure 5C). However, in contrast to the “MITF-low” GES [18], melanoma cells with the “EMT” GES showed high expression levels of EMT-TFs (*ZEB1*, *SNAI1*, and *PRRX1*), which have been demonstrated to play a critical role in melanoma progression [52,57]. In contrast, transcriptome classification of a large primary melanoma cohort yielded a GES characterized by high expression of both immune genes and EMT-related genes [24]. Furthermore, we identified a “Melanogenesis” GES that shared some characteristics with the previously identified “pigmentation” GES in metastatic [22] and primary [23] melanomas. It was enriched with the pigmentation biological process, had high expressions of melanogenesis-related genes, showed the highest expressions of *MITF*, and was related to poor prognosis. Melanogenesis is a melanoma-specific biological characteristic of cancers, and previous studies have reported its effect on melanoma prognosis [5,10,11,58]. Consistent with our findings, patients with melanotic and pigmented lymph node metastatic melanoma had a poorer prognosis than patients with amelanotic melanoma [11]. Moreover, when comparing high- and low-pigmentation tumors, shorter disease-free survival was observed in metastatic tumors with increasing pigmentation [11]. The relationship between radiotherapy efficiency and pigmentation was investigated in a previous report [10]. Patients with amelanotic melanoma showed a better prognosis than those with melanotic melanoma after chemo- and/or radiation treatment [10]. The “Melanogenesis” GES also showed some immune characteristics less comparable to the “Immune cell interactions” GES (Figure 2A), which may have been due to the cytotoxic and antiproliferative effects of melanogenesis mediators against T lymphocytes [5,9]. L-DOPA treatment for peripheral blood lymphocytes inhibited cell proliferation and significantly decreased the expression levels of the inflammatory cytokines (IL-1beta, IL-6, IL-10, and TNF-alpha) [9]. However, the inhibition of melanogenesis increased the susceptibility of melanoma cells to the cytotoxic effect of IL-2-activated immune cells [9].

Our analysis identified a newly reported GES named “Ribosomal biogenesis”, characterized by the high expression of ribosomal proteins and cell-cycle-related genes (Appendix A). Recent analyses have suggested that ribosomal biogenesis has a variety of functions beyond simply facilitating translation [59]. For example, ribosomal proteins have been reported to have cellular proliferation-, migration-, and invasion-related functions in addition to their role as ribosomal component proteins [60]. Interestingly, the “Ribosomal biogenesis” GES indicated the poorest prognosis compared with the other identified GESs (Figure 6). Numerous studies have consistently suggested that the tumorigenesis process may be fueled by ribosomal proteins, which can be used as a new target for cancer therapy [59,61].

Numerous studies have used differential gene expression analyses to compare metastatic and primary melanomas in order to identify prognostic markers [26,27,62,63]. Additionally, it has been reported that patients with low immune cell infiltration have poor clinical outcomes [27,64,65,66]. In our research, melanoma patients having “Ribosomal biogenesis” GES had the worst prognosis compared with the other identified GESs (Figure 6A,B,F) and showed a low level of immune cell enrichment (Figure 4D). Furthermore, melanoma cells having the “Ribosomal biogenesis” GES (cluster 4) showed weak cell–cell interactions with immune cells within MHC (I and II), and IFN-II pathways (Figure 3D,E,G). Therefore, we used the DEGs derived from this GES to construct our prognostic signature (MPS_45). The correlation of MPS_45 with prognosis was validated in several melanoma datasets (Figure 8) and was independent of the AJCC staging. A direct comparison of the genes included in our MPS_45 and previously reported prognostic signatures did not show many overlapping genes. Only *TYRP1* was shared with the previously reported Gerami signature [26], however, 16 genes (*AURKA*, *CCNB2*, *CDC20*/*A4*, *CENPF*, *CKS1B*, *KIF2C*, *KPNA2*, *LDHB*, *MAD2L1*, *MCM4*, *PRC1*, *TK1*, *TPX2*, *UBE2C*, and *ZWINT*) were shared between our MPS_45 and the established “high-grade” signature [23]. A previous pioneering study [49] reported an immune-resistant signature associated with immunotherapy resistance in patients with melanoma. This signature included 302 genes induced by and 292 genes repressed by malignant melanoma cells, correlated with T cell enrichment [49]. Our MPS_45 shared 19 genes (*CKS1B*, *EIF3K*, *IMPDH2*, *LDHB*, *LSM4*, *MCM7*, *MRPL37*, *NCL*, *NME2*, *RAN*, *RPL10*/*18*, *RPS19*/*3*, *RUVBL2*, *SAE1*, *TIMM13*/*50*, and *TUBA1B*) with the 302 induced genes.

Unlike previous studies [26,27,62,63,64], we did not consider DEGs between primary and metastatic or normal and tumor melanomas in our analysis. Alternatively, we applied our GES classification to primary and metastatic single melanoma cells simultaneously to focus on the biological characteristics of the melanoma cells and their relation to the clinical outcome. Moreover, instead of depending only on bulk RNA-seq analysis in identifying the GESs [18,22,23,24], in our analysis, we used the advantage of integrating scRNA-seq and bulk RNA-seq with prognostic data to identify the prognostic signatures. Considering the GES with the worst survival rate and weakest communication with immune cells, we identified our prognostic signature (MPS_45). Most intriguingly, the MPS_45 genes showed high expression levels not only in TCGA-SKCM patients with “Ribosomal biogenesis” GES but also in patients with “Melanogenesis” and “Extracellular structure organization” GESs (Figure 9B). These three GESs represented more than 58% of TCGA-SKCM patients and were the only GESs with poor prognostic characteristics. These findings together with overall survival analysis validations in several independent melanoma datasets, indicate the potential of our prognostic signature regardless of the biological state or clinicopathological stage of melanoma. The significance (*p*-value) of our identified prognostic signatures was better than or comparable to previous reports [22,23,24,27,56,67,68]. Additionally, the relatively small numbers of genes included in our MPS_45 increase its feasibility and ease in clinical applications. Further investigations of MPS_45 using the immunostaining of human tissue samples and in different treatment groups of melanoma patients are needed for better incorporation into clinical use.

## 5. Conclusions

In conclusion, we characterized the GESs of malignant melanoma patients at the single-cell level using scRNA-seq analysis. We found that melanoma cells are characterized by six distinct GESs (referred to as: “Anti-apoptosis”, “Immune cell interactions”, “Melanogenesis”, “Ribosomal biogenesis”, “Extracellular structure organization”, and “EMT”). Notably, the combined analysis of scRNA-seq and bulk RNA-seq data revealed that these six identified GESs were present predominantly in TCGA-SKCM patients. “Melanogenesis”, “Ribosomal biogenesis”, and “Extracellular structure organization” GESs can be used as prognosis indicators, and the “Immune cell interactions” GES can indicate improved survival in melanoma patients. Most importantly, using univariate, ridge, and multivariate Cox regression analyses, we uncovered and validated a 45-gene prognostic signature (MPS_45) as a potential predictor of malignant melanoma prognosis.

## Figures and Tables

**Figure 1 biomedicines-10-01478-f001:**
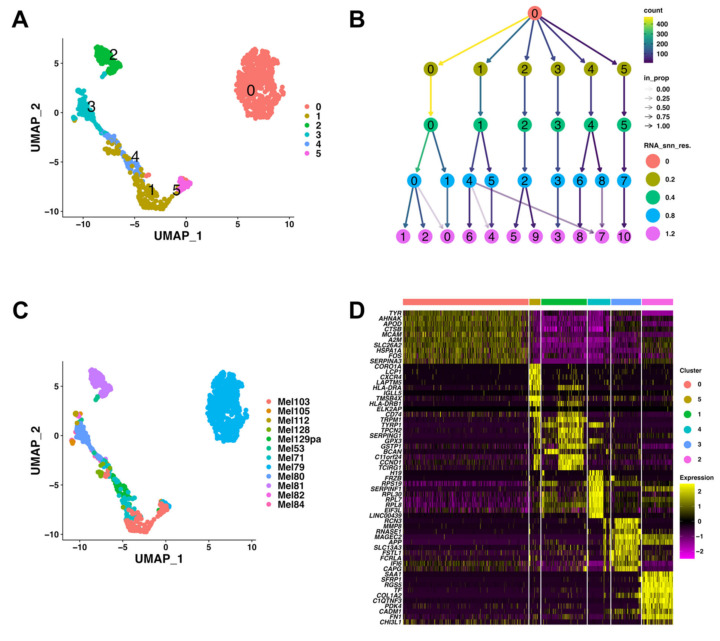
Melanoma patients exhibited different genetic profiles on the scRNA transcriptome level. (**A**) UMAP plot of 981 melanoma cells aggregated in six Louvain clusters. (**B**) Clustering tree for the melanoma cells clustered using various resolution parameters (0, 0.2, 0.4, 0.8, and 1.2). (**C**) UMAP plot showing patient sample distribution among the melanoma clusters. (**D**) The top 10 DEGs of each melanoma cell cluster are visualized by a heatmap.

**Figure 2 biomedicines-10-01478-f002:**
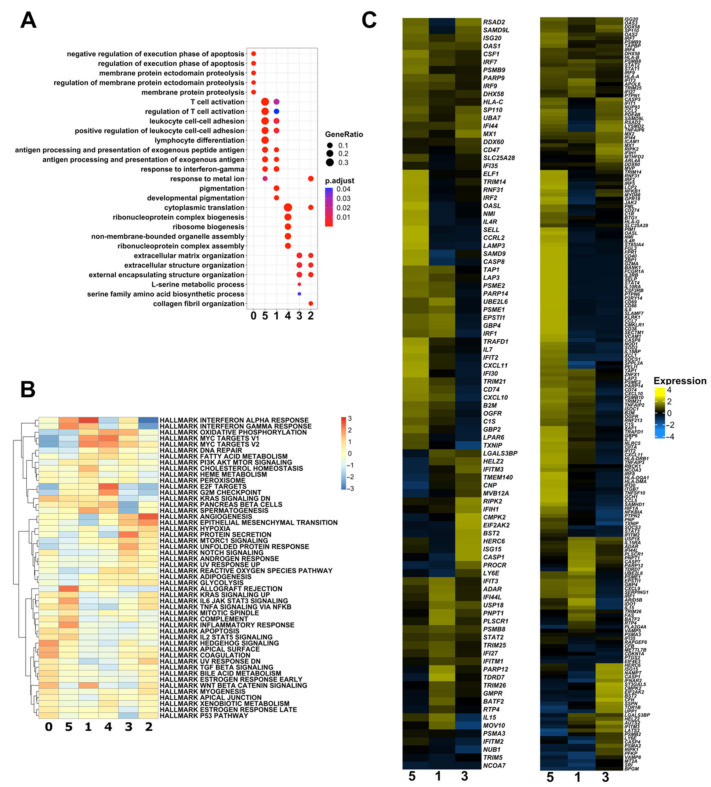
Melanoma cell clusters show unique characterization. (**A**) Dot plot representing the GO enrichment (biological process) for the DEGs of each melanoma cell cluster. (**B**) Heatmap visualization for the Hallmark GSEA in the melanoma cell clusters. (**C**) Heatmaps showing the expression levels of genes derived from HALLMARK_INTERFERON_ALPHA_RESPONSE gene set (left), and HALLMARK_INTERFERON_GAMMA_RESPONSE gene set (right).

**Figure 3 biomedicines-10-01478-f003:**
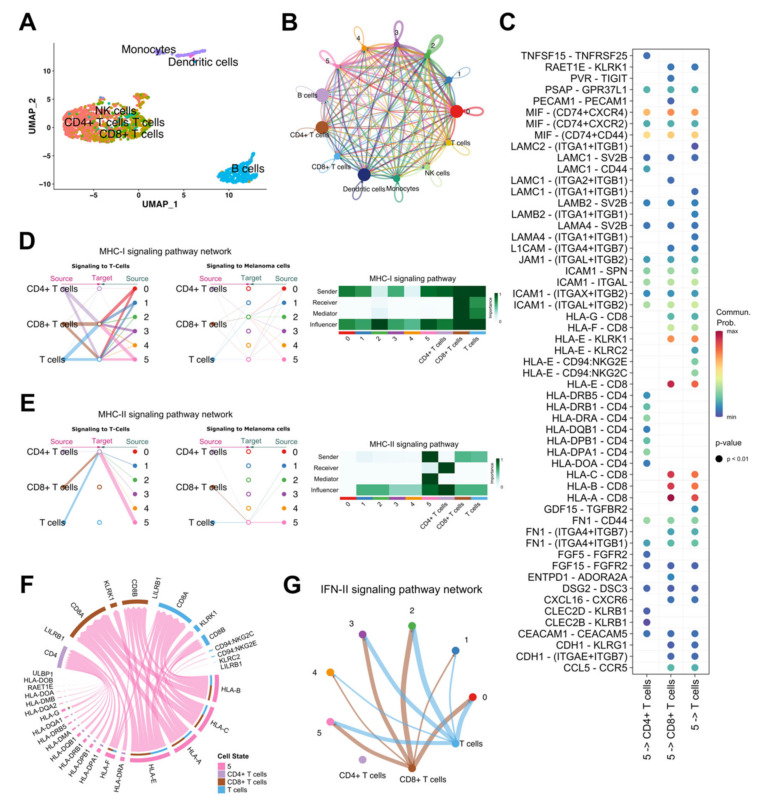
Cell–cell interaction between melanoma and immune cell clusters**.** (**A**) UMAP plot of aggregated immune cells from the GSE115978 dataset. Immune cells are automatically annotated with SingleR (R package), based on the Monaco Immune Database. Monocytes, dendritic cells, B cells, NK cells, T cells, CD8^+^ T cells, and CD4^+^ T cells could be identified. (**B**) Aggregated network of cell–cell communication, based on the total number of interactions. (**C**) Dot plot showing all the significant ligand–receptor interactions between cluster 5 and T cell subtypes (T cells, CD8^+^ T cells and CD4^+^ T cells). *p* < 0.01 was considered as the threshold for significance. (**D**) Hierarchical plot illustrates the communication network of the MHC-I signaling pathway (left). A heatmap showing the role (based on CellChat centrality measures) of each cell cluster in the MHC-I signaling pathway (right). (**E**) Hierarchical plot (left) and centrality heatmap (right) for MHC-II signaling pathway communication network. (**F**) Chord diagram showing ligand–receptor interactions between cluster 5 and T cell subtypes within MHC (I and II) signaling pathways only. (**G**) Circle plot for IFN-II signaling pathway.

**Figure 4 biomedicines-10-01478-f004:**
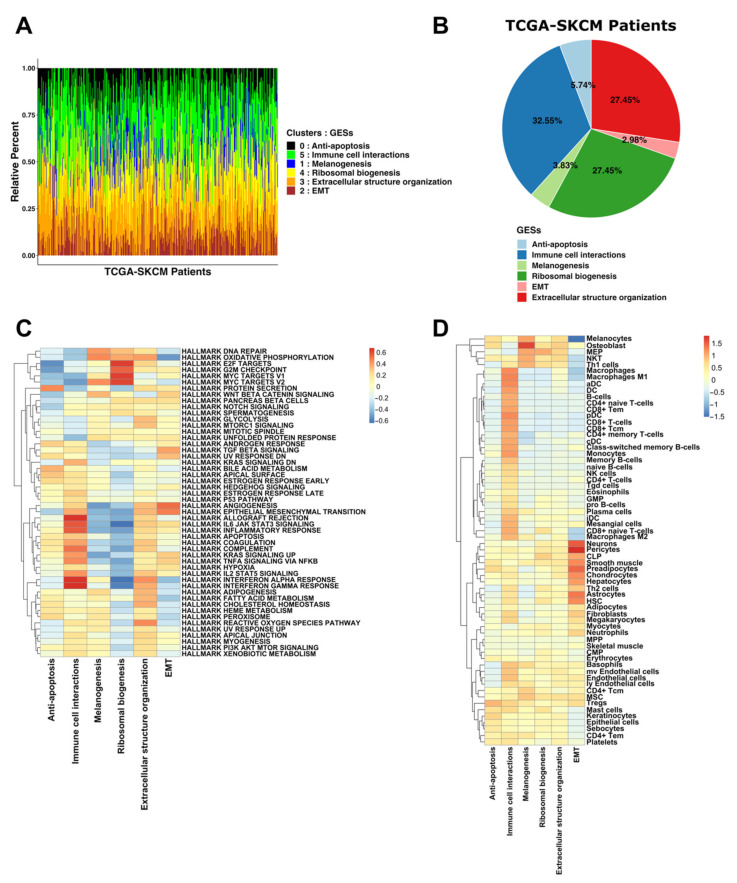
Classification of TCGA-SKCM patients based on the GESs determined by the scRNA-seq analysis. (**A**) Stacked bar chart showing the fractional compositions of the melanoma cells of the 6 scRNA melanoma cell clusters (GESs) in TCGA-SKCM samples. (**B**) A pie chart showing the fractions (expressed in percentages) of the melanoma GESs in the deconvoluted TCGA-SKCM samples. (**C**,**D**) Heatmaps showing the Hallmark GSEA (**C**) and cell-type enrichment (**D**) in the identified TCGA GESs.

**Figure 5 biomedicines-10-01478-f005:**
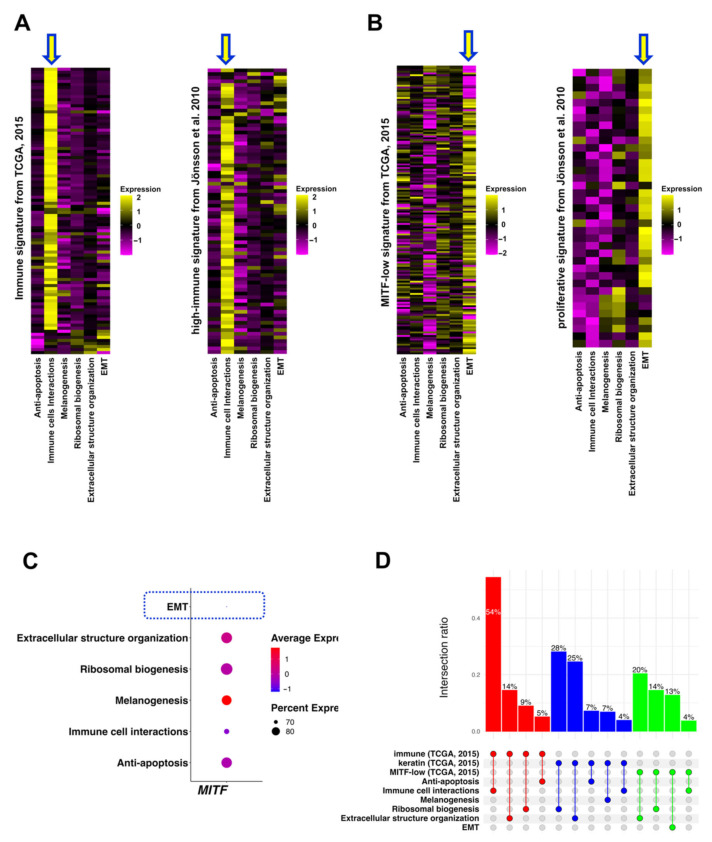
The identified GESs partially overlapped with previously reported GESs. (**A**) Heatmaps showing the expression of the representative genes of “immune” GES (left heatmap) and “high-immune” GES (right heatmap) in our identified TCGA GESs. Arrows indicating GESs with the highest similarity. (**B**) Heatmaps showing the expression of the representative genes of “MITF-low” GES (left heatmap) and “proliferative” GES (right heatmap) in our identified TCGA GESs. Arrows indicate GESs with the highest similarity. (**C**) Dot plot for the *MITF* expression levels in each GES. Dotted-line rectangle indicates the lowest expression level of *MITF* in “EMT” GES. (**D**) Upset plot visualizing the overlapping TCGA-SKCM patients between our GESs and previously reported GESs (“immune”, “keratin”, and “MITF-low”).

**Figure 6 biomedicines-10-01478-f006:**
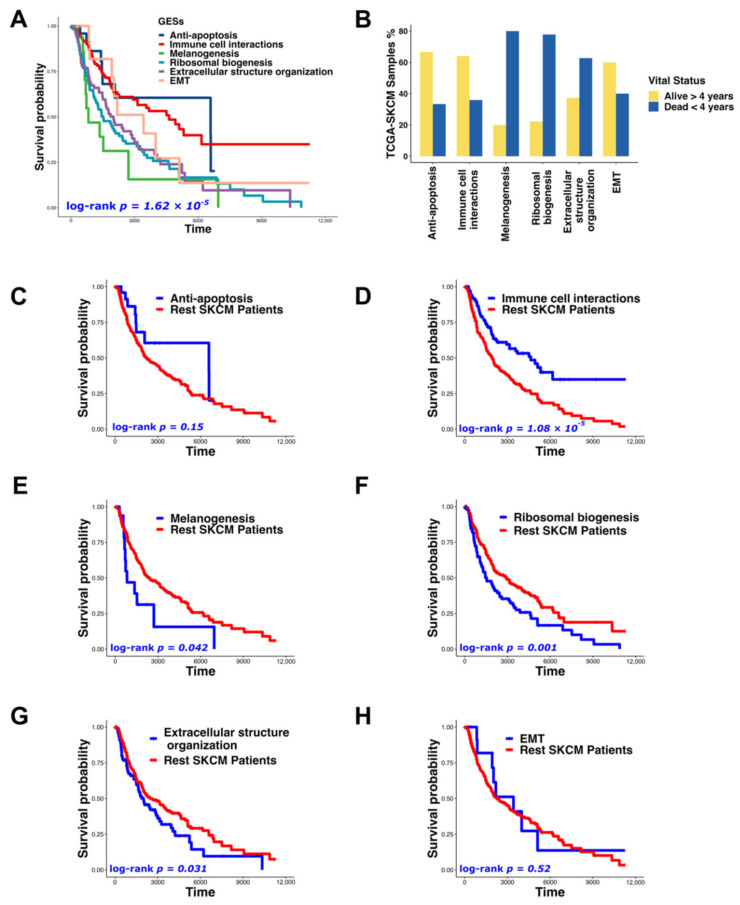
The identified GESs were correlated with melanoma prognosis. (**A**) Kaplan–Meier curve for the survival analysis of TCGA-SKCM patients with different GESs (log-rank *p* = 1.62 × 10^−5^). (**B**) A bar chart representing the proportion of dead (patients who died within the first 4 years) and alive (patients who were alive for more than 4 years) TCGA-SKCM patients with different GESs. (**C**–**H**) Kaplan–Meier survival curves comparing patients within each GES group ((**C**) “Anti-apoptosis”, (**D**) “Immune cell interactions”, (**E**) “Melanogenesis”, (**F**) “Ribosomal biogenesis”, (**G**) “Extracellular structure organization”, and (**H**) “EMT”) with the remaining TCGA-SKCM patients.

**Figure 7 biomedicines-10-01478-f007:**
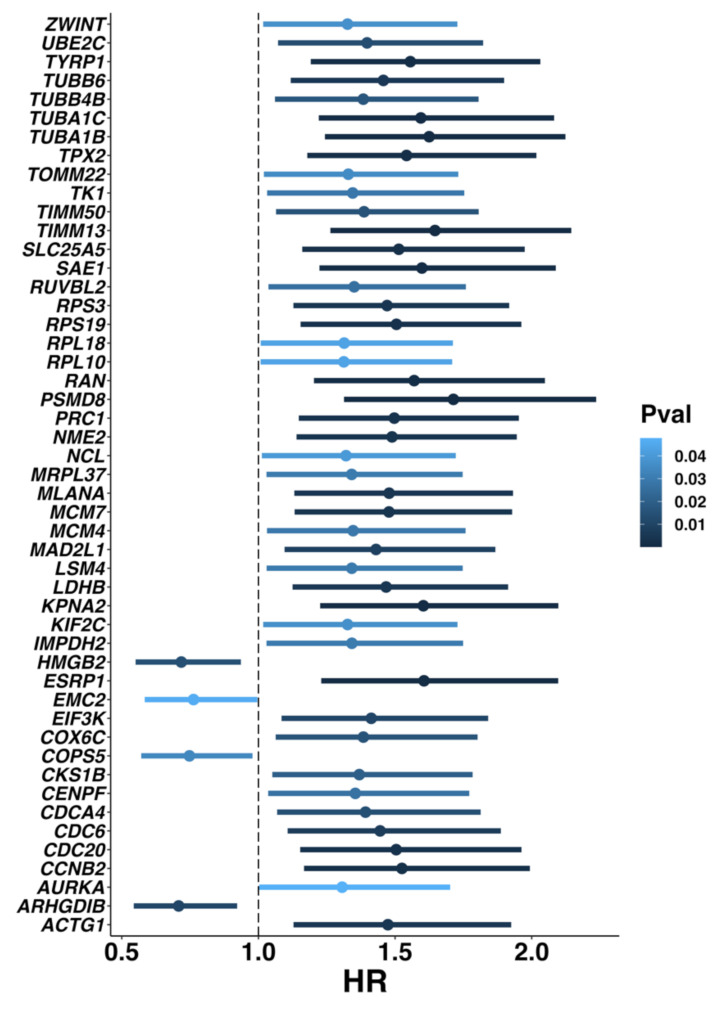
Univariate Cox regression analyses for DEGs of “Ribosomal biogenesis” GES. Forest plot for univariate Cox regression analyses of 49 out of 161 DEGs (|Log2FC| > 1, *p* < 0.05) which were significantly correlated with prognosis (*p* < 0.05). Only four DEGs (*HMGB2*, *EMC2*, *COPS5*, and *ARHGDIB*) had HR < 1, and the remaining DEGs had HR > 1.

**Figure 8 biomedicines-10-01478-f008:**
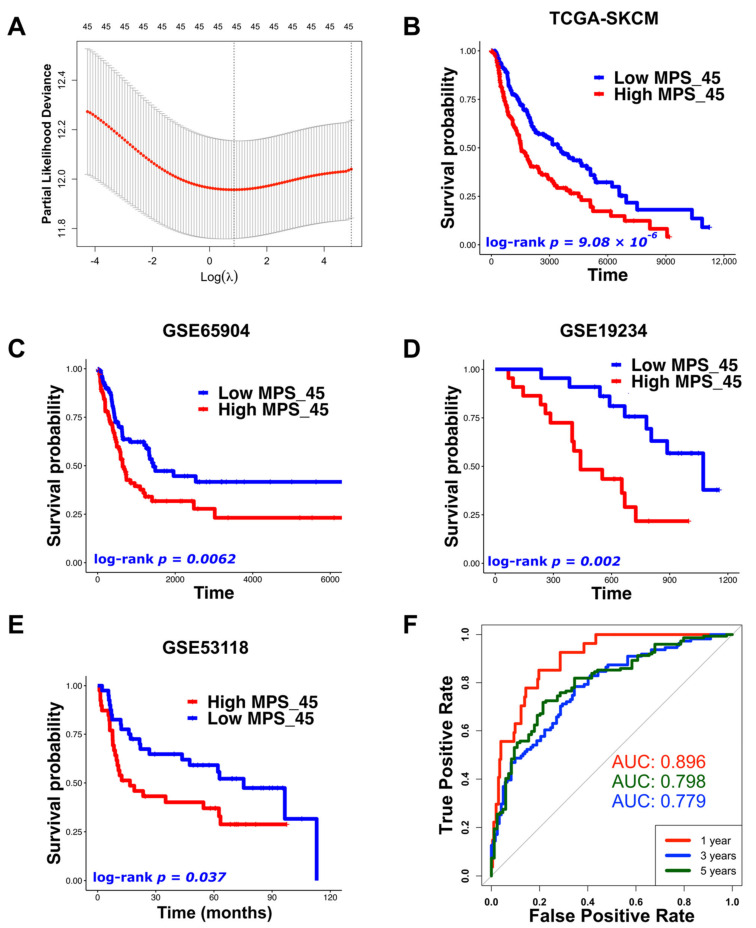
The prognostic signature (MPS_45) was significantly correlated with overall survival in TCGA-SKCM patients and external melanoma GEO datasets. (**A**) Ridge regression Cox analysis for the 45 DEGs of the “Ribosomal biogenesis” GES (alpha = 0). (**B**–**E**) Kaplan–Meier survival curves comparing high and low MPS_45 groups in (**B**) TCGA-SKCM (*p* = 9.08 × 10^−6^), (**C**) GSE65904 (*p* = 0.006), (**D**) GSE19234 (*p* = 0.002), and (**E**) GSE53118 (*p* = 0.037) datasets. (**F**) Time-dependent ROC curves of the MPS_45 using TCGA-SKCM samples for overall survival at 1, 3, and 5 years (AUC = 0.896, 0.779, and 0.798, respectively).

**Figure 9 biomedicines-10-01478-f009:**
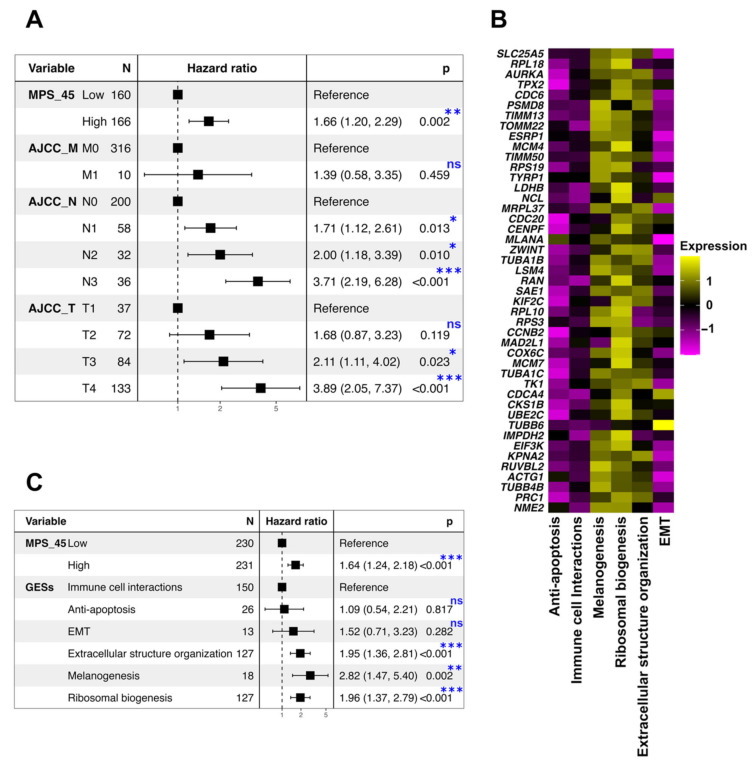
The MPS_45 was potentially associated with prognosis independent of the clinicopathological characteristics. (**A**,**C**) Forest plots for multivariate Cox regression analyses for the MPS_45 with (**A**) AJCC staging categories (M, N, and T) and with (**C**) the identified GESs groups. ns (non-significant) and * *p* < 0.05, ** *p* < 0.01, and *** *p* < 0.001. (**B**) Heatmap showing the expression levels of the MPS_45 genes in different TCGA-SKCM GES groups.

## Data Availability

The datasets analyzed during the current study are available in the GEO database (https://www.ncbi.nlm.nih.gov/geo/query/acc.cgi?acc=GSE115978 (accessed on 5 January 2022), https://www.ncbi.nlm.nih.gov/geo/query/acc.cgi?acc=GSE65904 (accessed on 15 September 2021), https://www.ncbi.nlm.nih.gov/geo/query/acc.cgi?acc=GSE19234 (accessed on 7 March 2022), and https://www.ncbi.nlm.nih.gov/geo/query/acc.cgi?acc=GSE53118 (accessed on 7 March 2022)), and TCGA (https://portal.gdc.cancer.gov/ (accessed on 14 December 2020)).

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
