# Peer review of "Analysis of Melanoma Gene Expression Signatures at the Single-Cell Level Uncovers 45-Gene Signature Related to Prognosis"

_biomedicines, 2022, doi:10.3390/biomedicines10071478_

Round 1
Reviewer 1 Report
The article entitled „Analysis of Melanoma Gene Expression Signatures at the Sin-2 gle-Cell Level Uncovers 45-Gene Signature Related to Prognosis” touches an important topic which is melanoma, characterized by one of the highest mortality rate among skin cancers. Most importantly, in the research prognosis indicators of the cancer were identified by means of analysing single-cell RNA sequencing data with statistical analysis. “Melanogenesis”, “Ribosomal biogenesis”, “Extracellular structure organization” were classified as gene expression signatures (GESs) with poor prognosis for the patients, whereas “Immune cell interactions” GES was linked to improvement of survival in patients. The work is interesting and the results may be used in the clinics in the future, however, a couple of minor suggestions may improve a bit the paper:
- Some unifications throughout the work are missing: in Fig. 8: I suggest p value should be presented with “<” sign or marked as ns (non-significant); interferon alpha/gamma (or: α/γ, respectively) should be used consistently instead of interchangeable use of interferon alpha/gamma and a/b.
- In line 308 there are too many points in the figure caption.
- The language should be revised and some punctuation corrections should be made (e.g. missing spaces after comas in the citations in the text or using “-“ instead of “–“ e.g. line 192, 195)
Author Response
Response to Reviewer 1 Comments
The article entitled „Analysis of Melanoma Gene Expression Signatures at the Sin-2 gle-Cell Level Uncovers 45-Gene Signature Related to Prognosis” touches an important topic which is melanoma, characterized by one of the highest mortality rate among skin cancers. Most importantly, in the research prognosis indicators of the cancer were identified by means of analysing single-cell RNA sequencing data with statistical analysis. “Melanogenesis”, “Ribosomal biogenesis”, “Extracellular structure organization” were classified as gene expression signatures (GESs) with poor prognosis for the patients, whereas “Immune cell interactions” GES was linked to improvement of survival in patients. The work is interesting and the results may be used in the clinics in the future, however, a couple of minor suggestions may improve a bit the paper:
We thank reviewer (1) for his/her support and belief in our research, and we are pleased that he/she found it interesting.
- Some unifications throughout the work are missing: in Fig. 8: I suggest p value should be presented with “<” sign or marked as ns (non-significant); interferon alpha/gamma (or: α/γ, respectively) should be used consistently instead of interchangeable use of interferon alpha/gamma and a/b.
Answer: We followed the reviewer's suggestion and used (ns) to refer to the “non-significant” p-value in Fig. 8 (it is now Fig. 9 due to exporting a new figure from the supplementary figures based on other reviewer suggestions) (page 18). We also added ns (non-significant) in the figure legend (page 18 - line 508). We unified interferon alpha/gamma through our manuscript (pages 5 and 6 - lines 260, 261, and 305:307).
- In line 308 there are too many points in the figure caption.
Answer: We removed extra points from the figure caption at line 308 and from the entire manuscript.
- The language should be revised and some punctuation corrections should be made (e.g. missing spaces after comas in the citations in the text or using “-“ instead of “–“ e.g. line 192, 195)
Answer: We revised the language and punctuation of our manuscript by ourselves and using the expert English editing services recommended by Biomedicines.

Reviewer 2 Report
This is timely and important paper on the bioinformatical analysis of pathways affecting melanoma behavior. However, there are deficiencies in referring to what is know in the literature, in particular as relates to melanogenesis and pigment cell biology, which lowered the grading of significance and scientific soundness to the average. These, can be easily corrected in the introduction and discussion.
I am surprised that the authors fail to discuss biochemical and physical properties of melanin and melanogenesis affecting melanoma behavior (Frontiers in Oncology 2022;12. DOI: 10.3389/fonc.2022.842496), isssues highly relevant to this manuscript.
Also the bioinformatic analyses should be discussed in the context of experimental data showing that melanogenesis affect cellular metabolism and local homeostasis (Arch Biochem Biophys 563:79-93, 2014; Pigment Cell Melanoma Res 25, 14-27, 2012; Anal Biochem 386:282-284, 2009). That melanogenesis has immunosuppressive effects (Int J Cancer 124, 1470-1477, 2009; Anticancer Res 18, 3709-3716, 1998). Also there are clinico-pathological data showing that melanogenesis affect outcome of advanced melanomas (Human Pathology42, 618–631, 2011) and response to the therapy (Oncotarget 20:17844-1785 2016 Feb 3. doi: 10.18632/oncotarget.7528)
The readers would also appreciate short information on different mechanisms regulating melanin pigmentation (Physiol Rev 84, 1155-1228, 2004; Pigment Cell Melanoma Res 25, 14-27, 2012).
Author Response
Response to Reviewer 2 Comments
This is timely and important paper on the bioinformatical analysis of pathways affecting melanoma behavior. However, there are deficiencies in referring to what is know in the literature, in particular as relates to melanogenesis and pigment cell biology, which lowered the grading of significance and scientific soundness to the average. These, can be easily corrected in the introduction and discussion.
We thank reviewer (2) for helping to improve the significance of our manuscript and providing us with exciting reference articles that impacted our manuscript's introduction and discussion style.
I am surprised that the authors fail to discuss biochemical and physical properties of melanin and melanogenesis affecting melanoma behavior (Frontiers in Oncology 2022;12. DOI: 10.3389/fonc.2022.842496), isssues highly relevant to this manuscript.
Also the bioinformatic analyses should be discussed in the context of experimental data showing that melanogenesis affect cellular metabolism and local homeostasis (Arch Biochem Biophys 563:79-93, 2014; Pigment Cell Melanoma Res 25, 14-27, 2012; Anal Biochem 386:282-284, 2009). That melanogenesis has immunosuppressive effects (Int J Cancer 124, 1470-1477, 2009; Anticancer Res 18, 3709-3716, 1998). Also there are clinico-pathological data showing that melanogenesis affect outcome of advanced melanomas (Human Pathology42, 618–631, 2011) and response to the therapy (Oncotarget 20:17844-1785 2016 Feb 3. doi: 10.18632/oncotarget.7528)
The readers would also appreciate short information on different mechanisms regulating melanin pigmentation (Physiol Rev 84, 1155-1228, 2004; Pigment Cell Melanoma Res 25, 14-27, 2012).
Answer: Based on the reviewer’s advice, we mentioned the effect of melanin pigment and melanogenesis on melanoma behavior in the introduction (page 1 - lines 31:46) section of our manuscript, referring to the research articles suggested by the reviewer (Front Oncol, 2022; Physiol Rev, 2004; Pigment Cell Melanoma Res, 2012; Arch Biochem Biophys, 2014; Int J Cancer, 2009; Oncotarget, 2016; Hum Pathol, 2013; Anal Biochem, 2009; Anticancer Res, 1998). Additionally, we enriched the discussion of our bioinformatics analysis with supporting experimental data from the papers proposed by the reviewer (page 19 - lines 687:703). Our analysis results prove the association of the “Melanogenesis” GES with poor prognosis and weak immune characteristics in patients with melanoma. We found it very helpful to confirm our results by referring to previous studies (Frontiers in Oncology, 2022; Int J Cancer, 2009; Oncotarget, 2016; and Human Pathology, 2011) (page 19 - lines 687:703). We also used another interesting study (Human Pathology, 2013) in our discussion.

Reviewer 3 Report
In this manuscript, Bakr et al used publicly available data to determine the prognosis-related signatures in melanoma. The authors used several bioinformatic and statistical analyses to dissect the different hallmarks of cancer and identified four gene signatures that were associated with prognosis in melanoma patients. However, the manuscript lack novelty and clarity.
Specific comments:
-
The characteristic of the tumour and their tumour microenvironment can also influence the prognosis of melanoma patients. The authors performed CIBERSORT and CellChat to evaluate the tumour immunobiology, and Figure S2 was provided to support the cell-cell interactions and their association with the gene signature. However, the results were not clear. The heatmap in Figure S2 requires simplification, for example, CD8A and CD8B can be stated as CD8 and choose one of the most important interacting ligands based on statistical significance. Figure S2 is the most interesting and distinguish this study from published work (GSE115978), however, it is embedded in the supplementary. The authors also did not provide enough results from CIBERSORT (e.g. proportion of immune cell infiltration).
-
The authors should perform TIMER or xCell to critically assess whether immune cells and their tumour microenvironment (stromal cells, fibroblast etc.) are impacting the prognosis of melanoma patients.
-
The Result and Discussion Sections are difficult to follow. Although the authors did perform all the analyses described in the Methods section, the results in the manuscript are a summary of what has already been reported in Jerby-Arnon Cell 2018 paper (GSE115978). Can the authors distinguish what new findings they have found as compared to the published study (GSE115978)?
-
The Discussion requires clarity. The authors stated “it has been reported that patients with low immune cell infiltration had poor clinical outcomes [18,53-55]. Accordingly, to exploit the prognostic characteristics of the “Ribosomal biogenesis” GES and its weak interactions with immune cells”. It is unclear to me how ribosomal biogenesis affects immune cells. Ribosomal biogenesis often refers to the translational capacity of the cell. Do the authors suggest that there are certain receptors/ligands were not translated properly and therefore weak interactions with the cells? Can the authors focus on the biology and correct the mis-interpretations of the biology in the Discussion?
- The study also lacks of immunohistochemistry experiment to support their scientific findings which were generated from bioinformatics.
Author Response
Response to Reviewer 3 Comments
In this manuscript, Bakr et al used publicly available data to determine the prognosis-related signatures in melanoma. The authors used several bioinformatic and statistical analyses to dissect the different hallmarks of cancer and identified four gene signatures that were associated with prognosis in melanoma patients. However, the manuscript lack novelty and clarity.
We appreciate reviewer (3)’s constructive criticism of our manuscript, which we found a great opportunity to improve our manuscript. In this revised manuscript, we want to clearly point out the main points of novelty. Firstly, starting from the scRNA-seq data of patients with melanoma, we identified six distinct gene signatures of melanoma cells, which could be retrieved in bulk RNA-seq data. Secondly, as mentioned by the reviewer, we have used several types of bioinformatics and statistical analysis to explain our identified signatures (ex: GO enrichment analysis, GESA of cancer hallmarks, cell–cell interaction, CIBERSORTx, and Cox regression analyses). Finally, besides recognizing four gene signatures associated with prognosis, we identified a 45-Gene prognostic signature. This 45-gene signature was validated for prognosis in several datasets and was significantly correlated with prognosis in melanoma patients (p-value < 0.05). The p-value is better than or comparable to previous reports (Jönsson et al., Clin Cancer Res. 2010; Harbst et al., Clin Cancer Res. 2012; Nsengimana et al., Oncotarget. 2015; Cirenajwis et al., Oncotarget. 2015; Lauss et al., J Invest Dermatol. 2016; Metri et al., Sci Rep. 2017; Thakur et al., Clin Cancer Res. 2019; Shou et al., Front Genet. 2020; Wan et al., Front Oncol. 2020). We believe that our approach utilizing scRNA-seq data and integrated analysis with bulk RNA-seq provides a valuable method of searching for new potential GESs or prognostic targets.
Specific comments:
- The characteristic of the tumour and their tumour microenvironment can also influence the prognosis of melanoma patients. The authors performed CIBERSORT and CellChat to evaluate the tumour immunobiology, and Figure S2 was provided to support the cell-cell interactions and their association with the gene signature. However, the results were not clear. The heatmap in Figure S2 requires simplification, for example, CD8A and CD8B can be stated as CD8 and choose one of the most important interacting ligands based on statistical significance. Figure S2 is the most interesting and distinguish this study from published work (GSE115978), however, it is embedded in the supplementary. The authors also did not provide enough results from CIBERSORT (e.g. proportion of immune cell infiltration).
Answer: We agree with the reviewer that Figure S2 is interesting, and unfortunately, we did not clearly explain it in the original manuscript. In the revised manuscript, we moved Figure S2 from the supplementary to the main text as Figure 3 (page 9), and we added further explanation about it under the results section (pages 5 and 6 - lines 257:298). In this analysis, we focused on the interaction between cluster 5 ("Immune cell interactions" GES) and T cells subtypes through Major Histocompatibility Complex (MHC) class I and II and type II interferon (IFN-II) signaling pathways based on the enrichment results (Figure 2, Table S3, and Table S4). Additionally, we simplified the heatmap in this figure, and following the reviewer's suggestion, we referred to CD8A and CD8B as CD8. We chose the highly significant ligand-receptor pathways to be visualized (page 9). In this study, we used the CIBERSORTx algorithm to explore the identified scRNA-seq GESs in bulk RNA-seq TCGA-SKCM data for our integrated analysis. Therefore, we did not analyze the proportions of cell types using this analysis method, instead, we used the xCell tool as suggested by the reviewer in the following comment, and showed new data (Figure 4D).
- The authors should perform TIMER or xCell to critically assess whether immune cells and their tumour microenvironment (stromal cells, fibroblast etc.) are impacting the prognosis of melanoma patients.
Answer: We want to express our gratitude to the reviewer for suggesting these analyses. We used the xCell tool to perform the cell-type enrichment analysis for the bulk RNA-seq data (page 4 - lines 171:172). Interestingly, the xCell results were complementary and aligned with the characteristics of our identified gene signatures (page 10 - lines 420:423). The “Immune cell interactions” GES, which we proved to have strong interactions with T cells (Figure 3) and to be correlated with improved survival (Figure 6A,B,D), was highly enriched with several types of immune cells. On the other hand, other GESs showed the attenuated enrichment of immune cells. Accordingly, we added a new panel to Figure 4 (Figure 4D) (page 11).
- The Result and Discussion Sections are difficult to follow. Although the authors did perform all the analyses described in the Methods section, the results in the manuscript are a summary of what has already been reported in Jerby-Arnon Cell 2018 paper (GSE115978). Can the authors distinguish what new findings they have found as compared to the published study (GSE115978)?
Answer: We want to thank the reviewer for raising such an important question, especially since we used the scRNA-seq data (GSE115978) obtained from Jerby-Arnon et al., Cell, 2018. However, we want to discuss essential differences and findings between our research and the GSE115978 study:
(1) Approach, objective, and study design: While Jerby-Arnon et al., Cell. 2018 focused on identifying an immune-resistant signature associated with immunotherapy resistance in melanoma patients, here in our study, we focused on exploring the characteristics of the melanoma cells and identifying as many gene signatures as possible (four gene signatures) related to prognosis and the biological characteristics of melanoma patients. That is why we included only data from untreated patients from the GSE115978 study.
(2) Main findings: The resistance program identified by Jerby-Arnon et al. was classified as induced (included 302 genes) or repressed (included 292 genes). Our analysis could identify prognosis-related four gene signatures, and we could retrieve and validate a prognostic signature that included 45 genes.
(3) The main advantage of our research is the small numbers of genes (45 genes) included in our identified prognostic signature, which makes it more applicable in clinical research.
We added new information in the discussion section to clearly state the significance of our findings compared with previous reports (page 20 - lines 766:782)
- The Discussion requires clarity. The authors stated “it has been reported that patients with low immune cell infiltration had poor clinical outcomes [18,53-55]. Accordingly, to exploit the prognostic characteristics of the “Ribosomal biogenesis” GES and its weak interactions with immune cells”. It is unclear to me how ribosomal biogenesis affects immune cells. Ribosomal biogenesis often refers to the translational capacity of the cell. Do the authors suggest that there are certain receptors/ligands were not translated properly and therefore weak interactions with the cells? Can the authors focus on the biology and correct the mis-interpretations of the biology in the Discussion?
Answer: We have revised the discussion section in our manuscript to explain the mentioned sentences more clearly (page 20 - lines 744:747). As an explanation, we meant to discuss the two main reasons for choosing the “Ribosomal biogenesis” GES to dig for our prognostic signature (MPS_45). These two reasons are: (1) the significant correlation with prognosis, as shown in bulk RNA-seq analysis (Figure 6A,B,F), and (2) weak cell–cell interactions between melanoma cells having this signature (cluster 4) and immune cells, predominantly within MHC class I and II and IFN-II signaling pathways (Figure 3D,E,G)
- The study also lacks of immunohistochemistry experiment to support their scientific findings which were generated from bioinformatics.
Answer: We agree with the reviewer about the importance of immunohistochemistry experiments for supporting our bioinformatics analyses. Our proposed 45-gene prognostic signature has to be validated using tissue samples from melanoma patients for clinical implementation. In this study, as we mentioned in regard to our novelty in the above response to the reviewer’s comment, we tried to demonstrate whether the integrated analysis of scRNA-seq and bulk RNA-seq data was able to search potential GESs. The validation of the clinical usability of the identified prognostic signature is our next topic of research. Therefore, we stated that it will be the focus of our future work in the discussion section (page 20 - lines 780:782).

Round 2
Reviewer 2 Report
The authors adequately revised the manuscript
Reviewer 3 Report
I don't have further comments.